# Analyzing the Effectiveness and Contribution of Each Axis of Tri-Axial Accelerometer Sensor for Accurate Activity Recognition

**DOI:** 10.3390/s20082216

**Published:** 2020-04-14

**Authors:** Abdul Rehman Javed, Muhammad Usman Sarwar, Suleman Khan, Celestine Iwendi, Mohit Mittal, Neeraj Kumar

**Affiliations:** 1National Center for Cyber Security, Air University, Islamabad 44000, Pakistan; abdurrahman.j74@gmail.com (A.R.J.); 171518@students.au.edu.pk (S.K.); 2Department of Computer Sciences, National University of Computer and Emerging Sciences, Islamabad 44000, Pakistan; m_usman296@hotmail.com; 3Department of Electronics BCC of Central South University of Forestry and Technology, Changsha 410004, China; celestine.iwendi@ieee.org; 4Department of Information Science and Engineering, Kyoto Sangyo University, Kyoto 603-8555, Japan; mohitmittal@ieee.org; 5Computer Science and Engineering, Thapar Institute of Engineering and Technology, Patiala, Punjab 147001, India

**Keywords:** activity recognition, smartphone, accelerometer sensor, smart health

## Abstract

Recognizing human physical activities from streaming smartphone sensor readings is essential for the successful realization of a smart environment. Physical activity recognition is one of the active research topics to provide users the adaptive services using smart devices. Existing physical activity recognition methods lack in providing fast and accurate recognition of activities. This paper proposes an approach to recognize physical activities using only2-axes of the smartphone accelerometer sensor. It also investigates the effectiveness and contribution of each axis of the accelerometer in the recognition of physical activities. To implement our approach, data of daily life activities are collected labeled using the accelerometer from 12 participants. Furthermore, three machine learning classifiers are implemented to train the model on the collected dataset and in predicting the activities. Our proposed approach provides more promising results compared to the existing techniques and presents a strong rationale behind the effectiveness and contribution of each axis of an accelerometer for activity recognition. To ensure the reliability of the model, we evaluate the proposed approach and observations on standard publicly available dataset WISDM also and provide a comparative analysis with state-of-the-art studies. The proposed approach achieved 93% weighted accuracy with Multilayer Perceptron (MLP) classifier, which is almost 13% higher than the existing methods.

## 1. Introduction

Physical inactivity is rising as a big issue nowadays. Authors in [1] present that inactivity is the 4th leading risk factor for people. Blood pressure and obesity are quite close to physical inactivity. Authors show that physical fitness can decrease mental disorder, cancer, diabetes, muscle issues, weight issues, emotional issues, and depression as well. Physical fitness can be tracked and analyzed by monitoring daily life physical activities.

Physical activity recognition was initiated back in 2004 using on-body sensors. Researchers in [2] used the accelerometer’s annotated data to recognize the physical activities. They made an Android-based system that collects raw data from the accelerometer and applied machine learning algorithms to predict physical activities. Authors in [3] recognized six basic activities, i.e., walking, jogging, sitting, standing, upstairs and downstairs. Authors in [4] used on-body sensors for activity recognition but found that it is very difficult to carry the sensors all the time. Many authors suggested that a smartphone is a non-obtrusive option for activity recognition [3,5,6,7,8,9].

The smartphone is playing a vital role in modern life. It provides services and applications such as health monitoring, early-stage disease detection, sports analysis, fitness tracking, and behavior analysis. Android-based smartphones have a built-in motion sensor that provides accurate and precise acceleration readings against physical activities. In early conditions, dedicated sensors were used for activity recognition. There exist several techniques for physical activity recognition such as on-body obtrusive and non-obtrusive sensors [10,11,12,13]. Non-obtrusive sensors are used in smart homes and smartphones. In smart homes, different motion and door sensors are installed at different locations and the primary objective is to recognize and assess activities but in smart homes, physical activities (i.e., running, cycling) cannot be performed due to the nature of activities.

The most widely used sensors for recording physical activities data using a smartphone are the accelerometer, gyroscope, and position sensor [8,14,15,16,17,18,19,20]. An accelerometer is capable of tracking activity readings to infer complex user motions, such as tilt, swing, or rotation.

Researchers showed that the accelerometer sensor is the most reliable and cheapest alternate of wearable sensors for physical activity recognition [19,21,22,23]. They showed that the accelerometer can also be used in combination with other sensors such as a gyroscope, light, proximity, barometer, linear acceleration and magnetometer sensor for better activity recognition. Furthermore, there is a large increase in the inventions of daily monitoring systems that can detect the user’s health, lifestyle, activities, behavior, and emotions [24,25,26]. Some sensors (i.e., GPS, Microphone, Radio-frequency and Near Field Communication) are also useful in detailed health monitoring [27,28,29,30].

Numerous authors have presented various approaches for activity recognition [3,5,6,7,8,9]. These studies mainly focuses on recognizing activities and the application of 3-axis of an accelerometer, which is not an optimal approach. However, our study presents an analysis of the effectiveness and contribution of each feature axis of an accelerometer in the recognition process and the use of only two axes (y and z-axis) of the accelerometer to provide fast and accurate recognition of daily life activities using machine learning models.

The key contributions of this research which also addresses the limitation of previous work can be summarized as follows:Investigates the effectiveness and contribution of each axis of the accelerometer for activity recognition.Proposes an approach to recognize the activities using only two axes of an accelerometer, resulting in fast and accurate recognition of physical activities.Presents an evaluation of variants of the sliding window.Uses three machine learning models and train their hyper-parameters for a more promising accuracy than state-of-the-art studies [3,8,15].In addition to our own collected dataset, the proposed approach was tested and evaluated extensively on a standard publicly available dataset WISDM [31] used in state-of-the-art studies.

The remainder of the paper is organized as follows. Section 2 explains the state-of-the-art studies related to physical activity recognition. Section 3 explains the proposed approach. Section 4 explains the experimental setup, the results, and demonstrates the analysis of each axis of an accelerometer. Finally, Section 5 concludes the work and details future directions.

## 2. Literature Review

Researchers have produced some frameworks and approaches that can recognize physical activities. Some of the activity recognition and monitoring frameworks is based on the smartwatches and multiple sensors worn on different body positions [4,32]. Some of these frameworks evaluate the classification accuracy of on-body sensor devices on different body positions [21,33,34,35]. Some frameworks estimate different time-domain features, sampling rates and analyze the trade-off between recognition accuracy and computational complexity [18]. The physical activity recognition with tri-axial sensors has gained more attention. By considering the effect of flexibility in orientation and placement, authors in [36] proposed an approach based on the coordinate transformation, PCA, and SVM. Authors in [33] proposed an approach for dynamic feature extraction and showed that feature selection is important for accurate recognition. They analyzed that Convolutional Neural Networks (CNN) gives better recognition results with dynamic features. In [37], the authors proposed an Ensemble Learning Machine (ELM) algorithm. According to the authors, ELM is the fastest algorithm to train for activity recognition and promising result. Authors in [18] proposed a methodology that utilizes data accumulated from an accelerometer to test it in a non-controlled condition. This investigation was particularly completed for old participants to improve their everyday life exercises. An older simulation unit was built that utilized three android cell phones that were put at the client’s midsection, knee, and lower leg.

Authors in [38] proposed a low-pass recognition system to recognize human activity. A set of five classifiers used was to classify physical activities using statistical features. Four participants were selected to collect the data at a rate of 100 samples per second (100/s) for 180–280 s. They selected a sliding window of 128 samples of each participant. They also showed different evaluation measures on a single classifier with a combination of other classifiers for further analysis. Authors in [39] used two accelerometers. They performed an offline evaluation of the dataset containing activities and different types of falls.

Researchers used a combination of sensors with an accelerometer for activity recognition. Authors in [4], used “eWatch” for data collection of six activities. Each “eWatch” device comprises of a bi-axial accelerometer and a light sensor. For training, they used different inductive learning models such as Decision Trees, k-Nearest Neighbor, Naive Bayes, and Bayes Net and evaluated their approach using five-fold cross-validation. Authors in [40,41,42,43,44,45,46] discussed the network protocol to use and process the sensors and wearable devices data over the network. The research [13] proposed an assisted approach that helps the participants to live healthily. The system recognizes physical activities (sleeping, walking and running) and suggests an optimal health care plan to participants with the help of doctors, guardians, and intelligent agent rankers.

In [33], the authors proposed an approach that extracts dynamic features from time-series human activity data by using recurrence plots [47]. They analyzed the dynamic vs. static features for activity recognition. First, the recurrence plots of each time series from gyroscope and accelerometer sensors were considered as dynamic features for activity recognition then Convolutional Neural Network (CNN) [48] was used for activity recognition. The study [49], present a smart-phone inertial sensors-based approach in which efficient features such as mean, auto-regressive coefficients, median, etc. extracted from raw data and then pass through Kernel Principal Component Analysis (KPCA) and Linear Discriminant Analysis (LDA) for dimension reduction. Finally, the Deep Belief Network (DBN) was used for activity recognition and was compared with Support Vector Machine (SVM) and Artificial Neural Network (ANN).

Although these studies used various approaches for activity recognition, they were not capable of recognizing a wide range of activities fast, accurately or efficiently. They also lacked battery limitation, better sliding window selection for recognition using machine learning model. Some studies used high-computational cost algorithms (i.e., Meta classifier, CNN and Random Forest) for recognition, which may result in a late response time when these systems apply to real-time recognition.

## 3. Proposed Methodology

In this section, we present our proposed approach for activity recognition. The proposed approach is comprised of pre-processing, feature extraction, data balancing, and recognition of activities. First, we make an Android application to collect readings from the accelerometer sensor. We collected data from twelve participants while performing six physical activities. Next, we pre-processed the raw data to remove the noise added at the start and end time while performing activities. Next, we extracted the features from the pre-processed data and make sliding windows of data. After that, we balanced all the activities so that one activity could not take advantage of its number of occurrences. Finally, the classifier was trained to recognize the activities. We used supervised machine learning classifiers i.e., Decision Tree (J48), Logistic Regression (LR) and Multilayer Perceptron (MLP) for physical activity recognition. The subsections below explain each part of the proposed approach. Figure 1 summarizes our proposed approach.

### 3.1. Data Collection

This section shows the rules and techniques to collect data for activity recognition. The data were collected using an Android application that runs on Android smartphones having an operating system greater than 4.0. There are three labels on the application interface showing the tri-axis (x-axis, y-axis, z-axis) readings of the accelerometer. Each axis returns a numeric value. Participants are asked to place their smartphone in the right pocket while performing all the activities.

### 3.2. Preproscessing

Initially, We set the data collection frequency to 50 ms, which provides 20 samples per second. After the axis analysis, as stated in Section 4.3, we observed that it is not an optimal way to set the frequency of accelerometer for all activities statically as it produces similar instances of (standing, walking) and (upstairs, downstairs) due to the indistinguishable nature of the activities. Due to this, we decreased the sampling frequency to 1 sample instance per one second for upstairs, downstairs, standing and sitting and 10 sample instances per second for the other two activities walking and jogging. We noticed that using this setting, it improves the battery timing as well as the recognition accuracy.

### 3.3. Feature Extraction

Next, we converted the data into the model window. Every window contains 200 examples. This chosen window is sufficient for extricating the outcomes as it is favored by different research works i.e., [3,21]. An accelerometer’s data consist of 3 float numeric axes (x, y, z) readings. All the identified exercises lie between the range of [−20, 20] for all pivots. Afterward, we utilized these readings to recognize six exercises (standing, sitting, upstairs, the first floor, strolling and running).

### 3.4. Data Balancing

We performed the SMOTE [50] data balancing method in which the minority class instances are oversampled by generating new synthetic instances to improve the representation of minority classes. To make a new instance, it calculates the distance between an original instance and the nearest neighbors. Then, it multiplies the new distance with the range between 0 and 1 and then, it is added into the original instance and thus, a new instance comes into existence. Below, we explain the example of generating synthetic examples.

Suppose a sample (1,2) and let (3,4) be its nearest neighbor. (1,2) is the sample for which k-nearest neighbors are being identified. (3,4) is one of its k-nearest neighbors.

Let
(1)$$\begin{array}{c} s1_{1}=1,\;s2_{1}=2\\ s1_{2}=3,\;s2_{2}=4 \end{array}$$

The new samples will be generated as
(2)$$\left(s1^{{}^{\prime }},s2^{{}^{\prime }}\right)=\left(1,2\right)+rand\left(0-1\right)*\left(3-1,4-2\right)$$
where rand(0–1) generates a random number between 0 and 1.

### 3.5. Activity Recognition

We provide a brief overview of the algorithms that we use in this paper to recognize physical activities in below section.

**Decision Tree (J48)**: J48 classifier builds the trees based on their information gain (IG) and entropy [51]. It compares the IG of all the features and split the tree with the feature having the best IG.
(3)$$\begin{array}{c} E=H\left(f\right)=I_{E}\left(k_{1},k_{2},\ldots ,k_{J}\right)=-\sum_{i=1}^{J}k_{i}\log_{2}^{}k_{i} \end{array}$$
where *k* is the probability of class *i*. H(feature) is the entropy that basically measures the degree of “impurity”.
(4)$$\begin{array}{c} IG\left(F_{i}\right)=H\left(C\right)-H\left(C|Fi\right) \end{array}$$
IG of a feature $F_{i}$ is calculated using Equation (Equation 4) where *C* represents different classification classes and $F_{i}$ are the different features in the dataset.**Logistic Regression (LR)** LR is used for classification by adding some parameters in linear regression algorithm like adding sigmoid function and threshold if the value is higher then 0.5 it will be yes and if the value of predicator is less then 0.5 it will go to no class [51]. The logistic regression algorithm uses the logistic function of fitting a straight line or hyperplane to squeeze the output of a linear equation between 0 and 1.The mathematical definition of logistic regression is given below:
(5)$$logistic\;\left(\eta \right)=\;\frac{1}{1+exp\left(-\eta \right)}$$
and it respected as shown in below Figure 2.This research modelled the affiliation between outcome and attributes with the linear equations in the linear regression classifier.
(6)$$O^{\wedge \left(i\right)}=\beta_{0}+\beta_{1}x_{1}{}^{\left(i\right)}+\ldots ,\beta_{n}x_{n}{}^{\left(i\right)}$$Logistic regression prefers the probability between 0 and 1 for classification, so we put the right side of the equation into logistic regression function. This emphasizes the output to assume values between 0 and 1.
(7)$$P\left(O^{\left(i\right)}=1\right)=\frac{1}{1+\exp\left(-\left(\beta_{0}+\beta_{1}x_{1}{}^{\left(i\right)}+\ldots ,\beta_{n}x_{n}{}^{\left(i\right)}\right)\right)}$$
(8)$$\log\left(\frac{P\left(y=1\right)}{1-P\left(y=1\right)}\right)=\log\left(\frac{P\left(y=1\right)}{P\left(y=0\right)}\right)=\beta_{0}+\beta_{1}x_{1}{}^{\left(i\right)}+\ldots ,\beta_{n}x_{n}{}^{\left(i\right)}$$The above equation represents the log function for logistic regression, also called the log odd function. This theorem shows that the concept of logistic regression is a linear model for log changes. If one of the $x_{j}$ attribute is changed by one element, prediction changes.
(9)$$\frac{P\left(y=1\right)}{1-P\left(y=1\right)}=odds=\exp\left(\beta_{0}+\beta_{1}x_{1}{}^{\left(i\right)}+\ldots ,\beta_{n}x_{n}{}^{\left(i\right)}\right)$$We then compare what happens if one of the feature values is increased by 1, yet we look at the ratio of the two projections rather than looking at the difference:
(10)$$\frac{odds_{x_{j+1}}}{odds}=\frac{\exp\left(\beta_{0}+\beta_{1}x_{1}+\ldots \beta_{j}\left(x_{j}+1\right)\ldots ,\beta_{n}x_{n}{}^{\left(i\right)}\right)}{\exp\left(\beta_{0}+\beta_{1}x_{1}+\ldots \beta_{j}x_{j}\ldots ,\beta_{n}x_{n}{}^{\left(i\right)}\right)}$$
(11)$$\frac{\exp\left(a\right)}{\exp\left(b\right)}=\exp\left(a-b\right)$$
(12)$$\frac{odds_{x_{j+1}}}{odds}=\exp\left(\beta_{j}\left(x_{j+1}\right)-\beta_{j}x_{j}\right)=\exp\left(\beta_{j}\right)$$Finally, a one-unit change in a feature changes the odds ratio by an exp factor ($\beta_{j}$). This can also be represented as a one-unit shift in $x_{j}$ increases the log odds ratio by the same weight factor. Many research studies interpreted the odds ratio because it is understood that thinking about something’s log () is difficult to comprehend.**Multilayer Neural Network (MLP)**: MLP is a feed-forward neural network, that maps the inputs to an appropriate set of outputs [51]. Typically, the network consists of an input layer, a hidden layer(s), and an output layer as shown in Figure 3. Given an input node $x_{i}$, the output of the hidden node $h_{j}$ is given as
(13)$$h_{j}=\phi_{1}+\left(\sum_{i=1}^{n}w_{ij}+\theta_{j}\right)$$
where $w_{i,j}$ represents the weight between the *i*th input and *j*th hidden node, and $\theta_{j}$ represents the bias value. In contrast, the output will be given as
(14)$$output=\phi_{2}+\left(\sum_{j=1}^{n}w_{jk}+\theta_{k}\right)$$The mapping of inputs to outputs is an iterative process, where in each iteration, weights $w_{i,j}$ are updated. One of the commonly used algorithm is the Back Propagation algorithm, which updates the weights using
(15)$$W_{ji}\left(t+1\right)=W_{ji}\left(t\right)-\epsilon \frac{\partial Ef}{W_{ji}}$$
the error between computed and desired output is used to update the weights.

## 4. Evaluation and Analysis

In this section, we explain our experiments and present the range of evaluation measures which we decided to use for experimentation purposes, then present and analyze our results. For experimental evaluation, we first collected labelled data and then made a sample window of 200 samples and then divided data into segments. We used J48, MLP, and LR to recognize the activities. We used the customized setting for tuning the parameter of each classifier. We applied three-fold cross-validation for all experiments. This works by leaving 1:3 part of data for testing and uses 2:3 part of data for training. To further evaluate the significance of our proposed approach, we used Leave-one-Subject-out (LOSO) cross-validation on WISDM [31] dataset. We compared the results of our approach with state-of-the-art study [15] using LOSO for a fair comparison. LOSO works by leaving one participant’s activities for testing and train the model on other participant’s activities. As there are 26 participants in WISDM [31] dataset, this cycle repeats for all participants 26 times randomly.

### 4.1. Evaluation Measures

We chose four evaluation measures: accuracy, recall, precision, and f-score to ensure the reliability of the model. Below, we illustrate terms that can be useful for evaluation measure analysis. The accuracy measure is calculated by TP (True Positive rate: correctly recognized samples) divided by N (all the samples of all activities). The recall measure is calculated by TP (True Positive rate: correctly recognized samples) divided by TP+FN (False Negative rate: samples wrongly recognized as other activities samples). The precision measure is calculated by TP divided by TP+FP (False Positive rate: samples of other activities wrongly recognized as one activity samples). F-score is computed as the harmonic mean of recall and precision.
(16)$$Accuracy=\frac{TP}{N}$$
(17)$$Recall=\frac{TP}{TP+FN}$$
(18)$$Precision=\frac{TP}{TP+FP}$$
(19)$$F-Score=2\times \;\frac{Precision\times Recall}{Precision+Recall}$$

### 4.2. Results

Table 1, Table 2, Table 3, Table 4 and Table 5 demonstrates the results of our activity recognition approach. Table 1 and Figure 4 demonstrate the percentage of activities that are correctly recognized. For standing and sitting activities, we achieved 5% more recognition rate using j48 than the LR. MLP achieve 2% better recognition rate than j48 for both activities. For downstairs activity, using j48, we achieve 13% more recognition rate than LR, while MLP achieves 6% more recognition rate than j48. For walking activity, MLP achieves 1% and 2% more recognition rate than LR and j48. For upstairs, using j48, we achieve 14% more recognition rate than LR, while MLP achieves 10% more recognition rate than j48. For jogging activity, using j48, we achieve 1% and 2% better recognition rate than LR and MLP.

Table 2 and Figure 5 demonstrate the recognition in percentage. As a conclusion using MLP, we achieve 87% accuracy which is 2% and 8% better than j48 and LR respectively. For further experiments, we only use MLP classifier for axis analysis Section 4.3 and comparative analysis Section 4.4 with the state-of-the-art.

Figure 6 shows the correctly classified activities concerning misclassified activities using the Decision Tree (j48) classifier. 19.5% of upstairs instances are misclassified as downstairs and 24.5% instances of downstairs are misclassified as upstairs. The jogging activity is misclassified 2.5% as walking, while the standing, sitting and walking are classified with minor confusions.

Figure 7 shows the correctly classified activities concerning misclassified activities using the Linear Regression (LG) classifier. 24.5% of upstairs instances are misclassified as downstairs and 24% of downstairs instances are misclassified as upstairs. The jogging activity is misclassified 3.7% as walking and 1.0 as standing, while the standing, sitting and walking are classified with minor confusions.

Figure 8 shows the correctly classified activities concerning misclassified activities using the Multilayer Perceptron (MLP) classifier. 15.5% of upstairs instances are misclassified as downstairs and 22.5% instances of downstairs are misclassified as upstairs. The jogging activity is misclassified 4.0% as walking and 2.0 as standing, while the standing, sitting and walking are classified with minor confusions. It is observed that MLP classifier is an optimal choice amongst other fro activity recognition.

### 4.3. Axis Analysis of Activities

In this section, we analyze the effectiveness and contribution of each axis of the tri-axial accelerometer sensor to improve the recognition rate. We choose six daily life activities: *upstairs, downstairs, sitting, walking, jogging, and standing* in this work. The experiment results of this analysis are carried through the three-fold cross-validation. These activities strongly depict the human physical routine. The x-axis in each graph depicts the horizontal movement [+x,−x] of the leg, y-axis depicts the motion in upward and downward directions [+y,−y] while z-axis depicts the forward and backward motion [+z,−z] of the leg.

The acceleration plots for the Six Activities (a–f) shown below in Figure 9 represents the acceleration and time of all the activities. Red curves are showing the x-axis of the accelerometer, blue lines are showing the y-axis of the accelerometer and the purple line are showing the z-axis of the accelerometer. The x-axis of the graphs shows the time in seconds, is accessed while the y-axis of the graph shows the acceleration of each sensor ranging between [−20,20]. Below we show the behavior of each activity according to each axis.

**Walking**: Figure 9a depicts that the periodic pattern of walking activity is quite similar to Figure 9b. While walking upward and downward periodic motion can be seen as light blue strides while the forward motion can be seen in purple strides and red lines showing horizontal movement is stable. The value of the x-axis lies between [−12,12], y-axis value [−18,5] & the value of z-axis lies between [−9,16] which is quite unique. These observations can be proven by Table 3 and Figure 10 depicting a combination of the different axis of the accelerometer to recognize the activities. In Table 3 the recognition rate of walking activity against x−y axis is 89.5%, x−z axis is 90.0%, y−z axis is 95.0%, and x−y−z is 92.5%. Hence, for walking activity, the y-axis and z-axis of the accelerometer sensor are the potential features.**Jogging**: Figure 9b depicts periodic patterns quite similar to the Figure 9a. Upward and downward motion can be seen as light blue strides while the forward motion can be seen in purple strides which are quite high than walking and red lines showing horizontal movements are stable. In Table 3, it is shown that the recognition rate of jogging activity against x−y axis is 92.2%, x−z axis is 94.1%, y−z axis is 96.5%, and x−y−z is 94.0%. Hence for this activity y-axis and z-axis of the accelerometer sensor are the potential features. By looking at accelerometers detected readings, it shows that the range of x-axis falls between [−8,10], y-axis value [−20,18] and the value of z-axis lies between [−10,14]. In the previous case, the same type of results are obtained which further strengthen our observations.**Upstairs**: The patterns in Figure 9c are quite similar to the Figure 9b of jogging. While going upstairs, upward and downward motion can be seen as light blue strides while the forward motion can be seen in purple strides and red lines showing horizontal movements are stable. This activity is getting confused with jogging activity as the only difference between both is of the z-axis. There is a high upward positive peak and downward negative peak after a specific periodic interval which depicts that a user stepped up a stair and then suddenly it goes to the initial stage. The range of x-axis lies between [−12,10], y-axis between [−4,16] and z-axis lies between [−11,13], which is quite similar to the jogging. In Table 3, it is shown that the recognition rate of upstairs activity against x−y axis is 79.6%, x−z axis is 65.5%, y−z axis is 89.8%, and x−y−z is 79.3%. Hence, for this activity, y-axis and z-axis of the accelerometer sensor are the potential features.**Downstairs**: Figure 9d depicts that the periodic patterns are slightly similar to the Figure 9b. Each small peak represents movement down a single stair. The z-axis values show a similar pattern showing negative readings, reflecting the regular movement down each stair which was positive in the previous case (upstairs). While walking downstairs, downward motion can be seen as light blue strides which show a negative axis while the forward motion can be seen in purple strides and red lines, showing horizontal movements that are stable in the start but suddenly it shows a high read peak which turns a mentioned earlier in the dataset collection section. The range of x-axis falls between [−11,13], y-axis between [−12,−3] and the range of z-axis lies between [−10,15], which is quite similar to the Upstairs. In Table 3, it is shown that the recognition rate of upstairs activity against x−y axis is 70.7%, x−z axis is 68.5%, y−z axis is 87.1%, and x−y−z is 71.5%. Hence for this activity y-axis and z-axis of the accelerometer sensor are the potential features.**Sitting**: Figure 9e depicts that all reading against each axis is, constant and stable patterns without any regular periodic pattern. This activity shows a unique behavior. By looking at accelerometers detected outputs, we observe that the range of x-axis lies between [−1,0], y-axis value [−1,1] and the value of z-axis lies between [9,11]. In Table 3, it is shown that the recognition rate of upstairs activity against x−y axis is 90.2%, x−z axis is 86.8%, y−z axis is 94.7%, and x−y−z is 9.0%. Hence for this activity, y-axis and z-axis of the accelerometer sensor are the potential features.**Standing**: This activity also shows a unique behavior as depicted in Figure 9f. It is seen that accelerometer readings are constant and depicts stable patterns without any regular periodic pattern. The range of x-axis lies between [0,2], y-axis value [−9,−11] and the value of z-axis lies between [−1,1]. In Table 3, it is shown that the recognition rate of upstairs activity against x−y axis is 90.1%, x−z axis is 88.2%, y−z axis is 96.4%, and x−y−z is 93.7%. Hence, for this activity y-axis and z-axis of the accelerometer sensor are the potential features.

Besides our approach, authors in [52] used principal component analysis (PCA) for feature selection from fall detection datasets. This dataset contains 3 accelerometer Axis as features. According to the authors, PCA selects optimal features from the feature matrix. PCA measures the importance of a feature based on variance. Feature having high variance are treated as principal components and low variance feature are considered as noise. In this study, we focus on daily life six activities presented in Section 4.3. Applying PCA on our dataset provides the X-axis and Y-axis of the accelerometer as a principal component but provides an overall accuracy of 85.4% which is far less than our approach as shown in Table 3. The rationale behind this is that activities such as standing, sitting, walking and jogging can be recognized efficiently using x and y-axis but not other complex activities such as upstairs and downstairs. These activities are efficiently recognized with the accuracy of 93.3% using y and z-axis as shown in Table 3.

To sum up all the discussion, observations and analysis, we strongly claim to use the y-axis and z-axis of the accelerometer sensor to recognize activities: *Walking, Standing, Sitting, Upstairs, Downstairs and Jogging* experimented extensively in this paper.

### 4.4. Comparative Analysis

This section presents the results obtained by our approach in comparison with state-of-the-art approaches [3,8,15]. For a fair comparison, we also use the WISDM dataset [31]. This dataset was collected using a smartphone accelerometer from 36 participants. The participants were asked to carry a smartphone in their front pant’s leg pocket while performing activities as we collect in this work. The participants were asked to perform walking, walking upstairs, jogging, walking downstairs, sitting, and standing activities. This dataset is also used by various recent studies to improve activity recognition [3,5,6,7,8,15]. The axis analysis results in Table 3 and Figure 10 show that our approach achieves a promising recognition rate using the y-z axis. We also use the (y-z) axis data of the WISDM dataset to compare the results.

The Table 4 and Table 5 and Figure 11 demonstrate the recognition rates on each activity using proposed approach and state-of-the-art studies. In Table 4, for standing activity, we achieve 1% and 4% better recognition rate than [8] and [3] respectively. For sitting activity, we achieve same recognition rate as [8]. However, we achieve 1% less recognition rate than the study [3] on sitting activity. In case of downstairs activity, we achieve 17% and 47% better recognition rate than [8] and [3]. For walking activity, we achieve 10% and 5% better recognition rate than [8] and [3]. In case of upstairs activity, we achieve 27% and 23% better recognition rate than [8] and [3] respectively. However, we achieve 1% and 2% less recognition rate than the study [15] and [3] on jogging activity. By using (y-z) axis of WISDM dataset, we achieve 93% recognition rate which is 12% and 13% better recognition rate than [8] and [3] respectively.

The Table 5 shows the comparison results using Leave-One-Subject-Out (LOSO) cross-validation. It shows that for standing activity, we achieve the same recognition rate as [15]. For sitting activity, we achieve an 8% better recognition rate than [15]. In the case of downstairs activity, we achieve a 2% better recognition rate than [15]. However, for walking activity, we achieve a 3% less recognition rate than [15]. In the case of upstairs activity, we achieve a 10% better recognition rate than [15]. However, we achieve a 2% less recognition rate than the study [15] on jogging activity. By using (y-z) axis of WISDM dataset, we achieve 91% recognition rate which is 3% better than [15].

## 5. Conclusions and Future Work

In this paper, an approach is proposed to recognize physical activities using only 2-axes of the smartphone accelerometer sensor. The daily life activities data is collected and labeled using the accelerometer from 12 participants.The experimental results show that the proposed method is practical and capable of increasing the recognition rate of physical activities. Our approach achieved an accuracy of 93% and examined the role of each axis of an accelerometer to improve the recognition rate as shown in Table 4 and Table 5. Hence, the proposed approach provides more promising results compared to the existing techniques and presents a strong rationale behind the effectiveness and contribution of each axis of an accelerometer for activity recognition. The impact of the proposed approach and findings are significant for activity recognition. In the future, we would like to analyze more daily life activities using our technique.

## Figures and Tables

**Figure 1 sensors-20-02216-f001:**
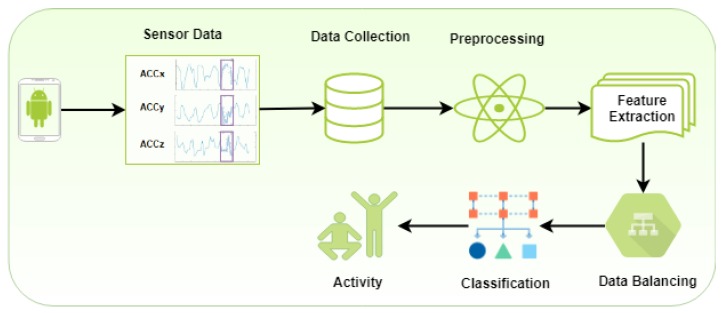
Block diagram of the proposed activity recognition approach.

**Figure 2 sensors-20-02216-f002:**
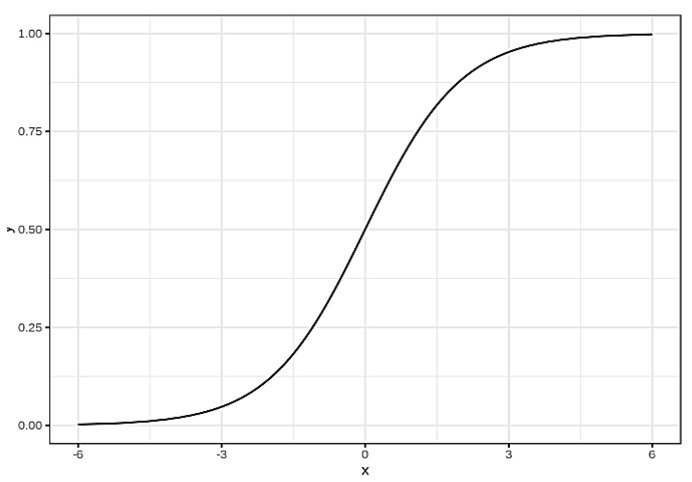
Illustration of LR classifier.

**Figure 3 sensors-20-02216-f003:**
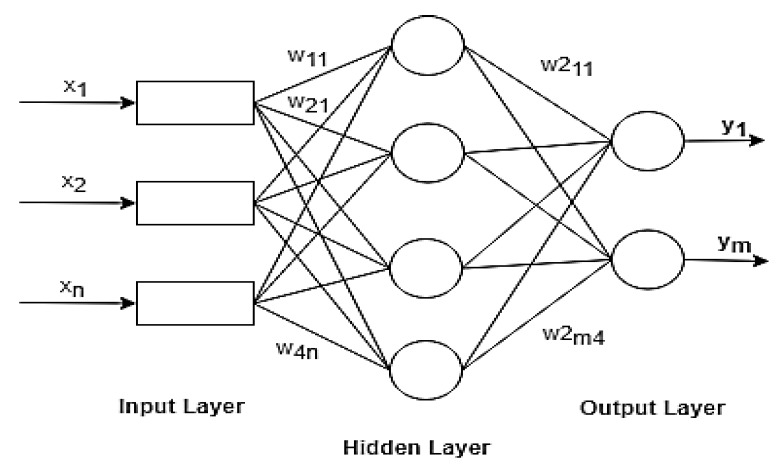
Structure of MLP classifier.

**Figure 4 sensors-20-02216-f004:**
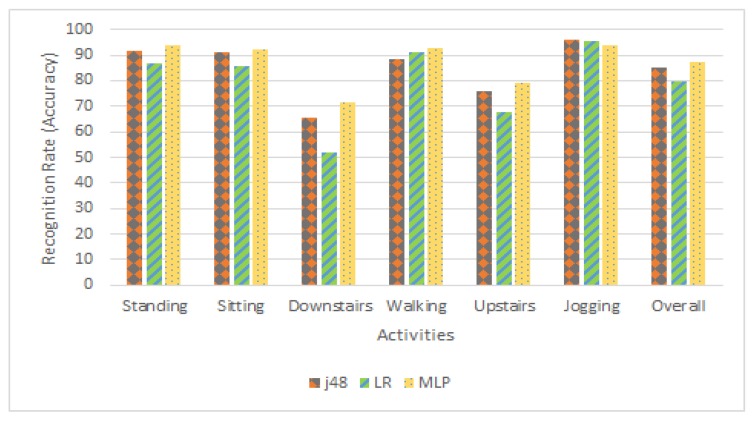
Recognition rate comparison of j48, LR and MLP classifier concerning each activity.

**Figure 5 sensors-20-02216-f005:**
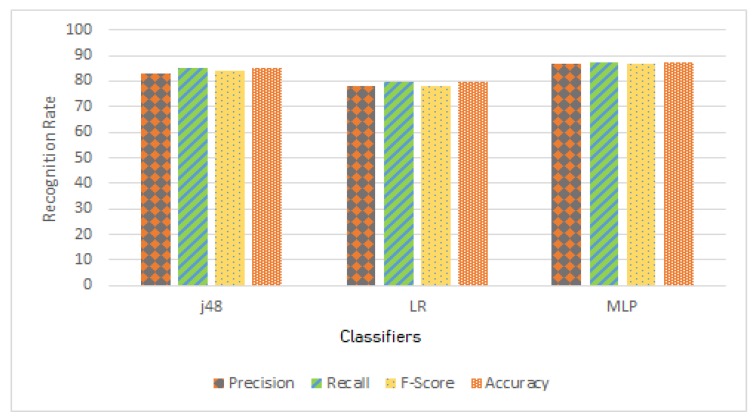
Comparison of evaluation measures with respect to j48, LR and MLP classifier.

**Figure 6 sensors-20-02216-f006:**
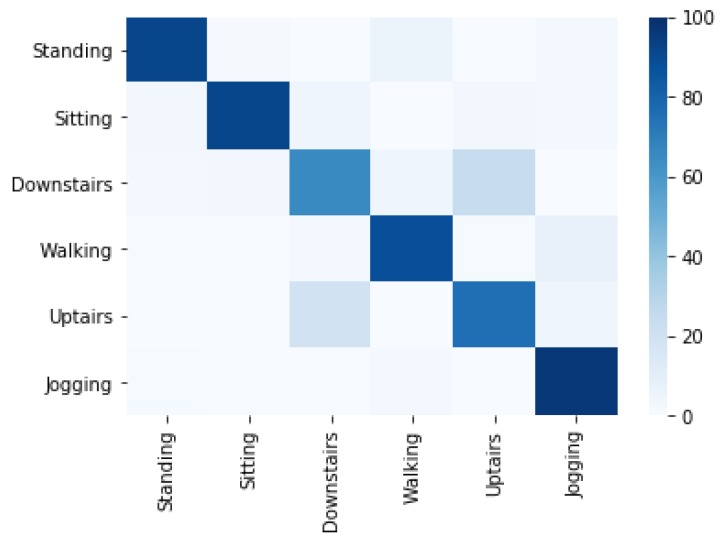
Confusion matrix of activity recognition using j48 classifier.

**Figure 7 sensors-20-02216-f007:**
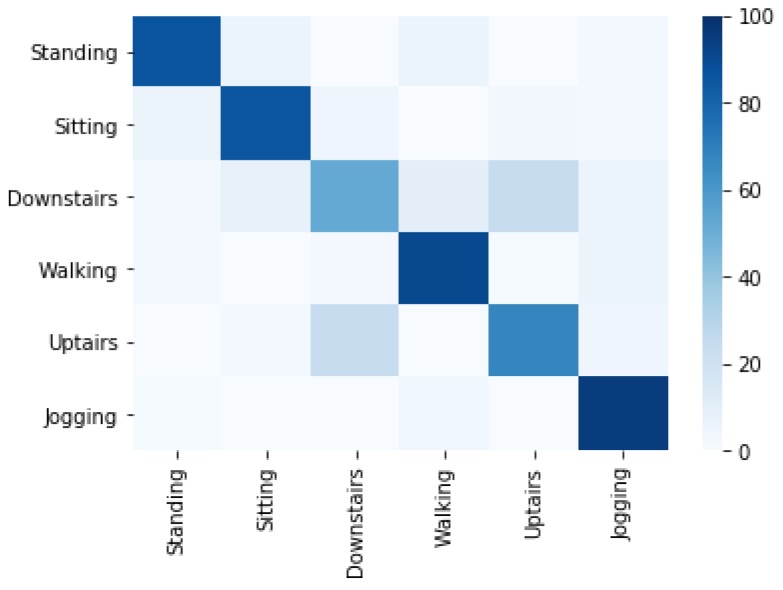
Confusion matrix of activity recognition using LG classifier.

**Figure 8 sensors-20-02216-f008:**
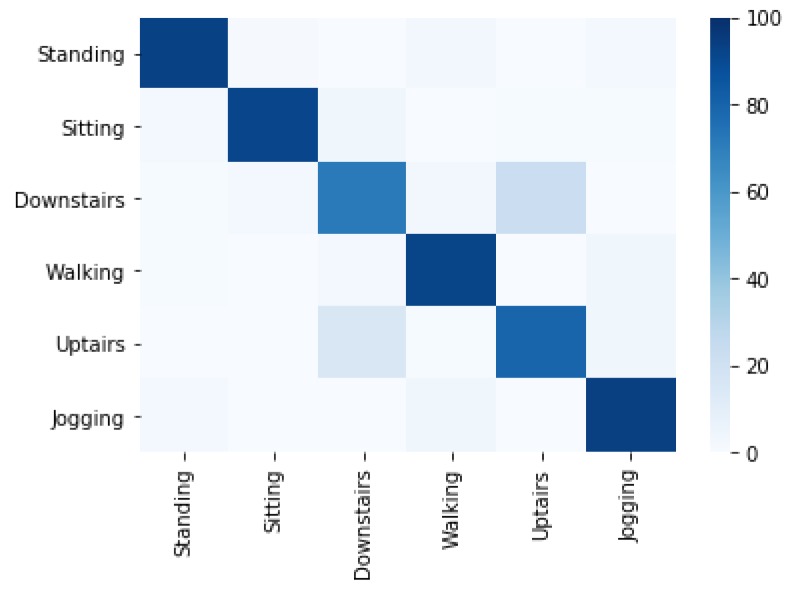
Confusion matrix of activity recognition using MLP classifier.

**Figure 9 sensors-20-02216-f009:**
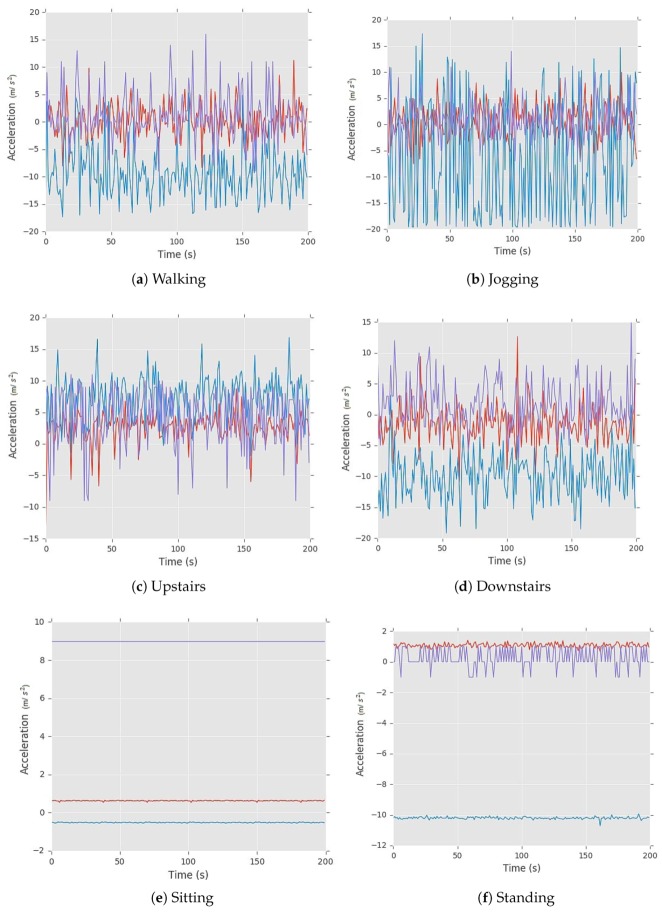
Acceleration plots of daily life physical activities.

**Figure 10 sensors-20-02216-f010:**
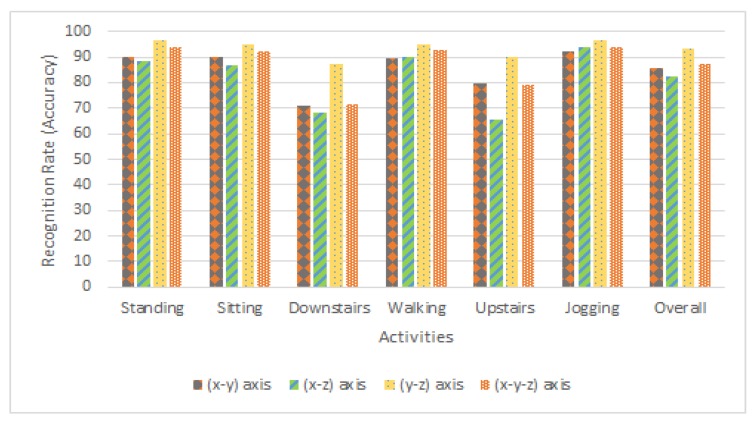
Recognition rate comparison of different combination of accelerometer axis with respect to each activity using MLP.

**Figure 11 sensors-20-02216-f011:**
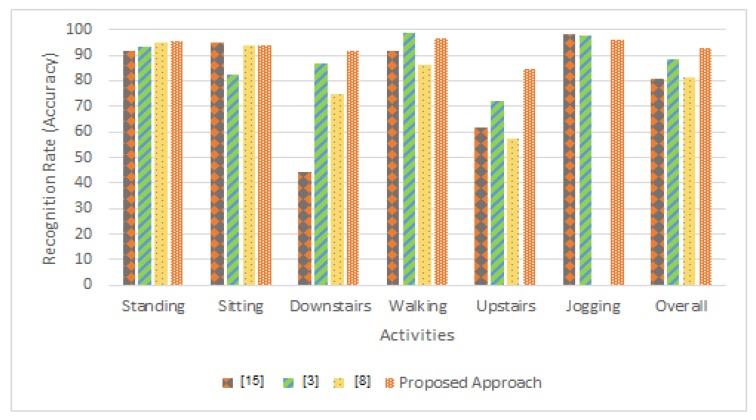
Comparison results of the proposed approach with the state-of-the-art research works.

**Table 1 sensors-20-02216-t001:** Recognition rate comparison of j48, LR and MLP classifier with respect to each activity.

	j48	LR	MLP
Standing	91.5	86.6	93.7
Sitting	91.2	85.5	92.0
Downstairs	65.7	52.1	71.5
Walking	88.3	91.0	92.5
Upstairs	75.6	67.5	79.3
Jogging	96.2	95.3	94.0
Overall	85.0	79.6	87.2

**Table 2 sensors-20-02216-t002:** Comparison of evaluation measures with respect to j48, LR and MLP classifier.

	Precision	Recall	F-Score	Accuracy
j48	83.2	85.0	84.0	85.0
LR	78.1	79.6	78.1	79.6
MLP	86.7	87.2	86.9	87.2

**Table 3 sensors-20-02216-t003:** Recognition rate comparison of different combination of accelerometer axis with respect to each activity using MLP.

	(x-y) axis	(x-z) axis	(y-z) axis	(x-y-z) axis
Standing	90.1	88.2	**96.4**	93.7
Sitting	90.2	86.8	**94.7**	92.0
Downstairs	70.7	68.5	**87.1**	71.5
Walking	89.5	90.0	**95.0**	92.5
Upstairs	79.6	65.5	**89.8**	79.3
Jogging	92.2	94.1	**96.5**	94.0
Overall	85.4	82.2	**93.3**	87.2

**Table 4 sensors-20-02216-t004:** Comparison results of the proposed approach with state-of-the-art research works.

Activities	[3]	[8]	Proposed Approach
Standing	91.9	94.9	**95.7**
Sitting	95.0	93.9	**94.0**
Downstairs	44.3	74.6	**91.5**
Walking	91.7	86.3	**96.5**
Upstairs	61.5	57.2	**84.3**
Jogging	98.3	N/A	**96.0**
Overall	80.7	81.4	**93.0**

**Table 5 sensors-20-02216-t005:** Comparison results of the proposed approach with state-of-the-art research work [15] using leave-one-subject-out cross-validation.

Activities	[15]	Proposed Approach
Standing	93.3	**93.2**
Sitting	82.6	**90.3**
Downstairs	87.0	**89.1**
Walking	98.5	**95.9**
Upstairs	72.2	**82.8**
Jogging	97.8	**95.5**
Overall	88.5	**91.13**

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
