# Peer review of "Analyzing the Effectiveness and Contribution of Each Axis of Tri-Axial Accelerometer Sensor for Accurate Activity Recognition"

_sensors, 2020, doi:10.3390/s20082216_

Round 1

Reviewer 1 Report

The authors focused on recognizing the typical activities of human beings by using a three-axis accelerator. The work is interesting but some problems should be clarified and improved.

  1. The authors should highlight the innovation and contribution of this work.
  2. The introduce of proposed method is not clear. The author should emphasize the unique of proposed method and the reason of its effectiveness rather than the basic knowledge of classification algorithms.
  3. The full name and definition of the abbreviations in Eq.(14)-(17) should be given for the readers who don’t work in the field of machine learning.
  4. The explanation of “4.2 Dataset” is very confusing. It seems “Ai” is a specified activity (i.e. the label of the sensor signals). Why is “As” defined by the accelerometer sensor and what is “t”?
  5. The percentage of data for the training and testing should be clarified. And, the recognition rates in the section of 4.3 is the training result, or the testing result?
  6. The name of X axis in Fig. 5 should not be “activities”.
  7. What ‘s the unit of the acceleration shown in Fig. 9?
  8. The definition of XYZ should be given before discussing the effect of the specified direction for the detected accelerations.
  9. Why could the proposed method have the better result in WISDM dataset?
  10. There are some grammar and typo errors, e.g., the missing of articles and “fro” in L212 of P9.

Author Response

`Revision of Manuscript JManuscript ID: sensors-759417

Analyzing the Effectiveness and Contribution of each Axis of Tri-Axial

Accelerometer Sensor for Accurate Activity Recognition

Submitted to Sensors

We would like to thank the reviewers and Editor-in-Chief for their valuable comments on the manuscript along with their suggestions of improvements. These comments and suggestions have been considered when preparing the revised version of the manuscript. The remainder of this response letter explains how we have handled the reviewers’ comments and implemented their suggestions. We hope that the changes and improvements that we have made in the manuscript will satisfy the requirements of both the reviewers and the Editor-in-Chief, which will lead to accepting the manuscript for publication in a forthcoming issue of Journal of Parallel and Distributed Computing.

Editor in Chief’s initial decision: Major revision

Response: We thank the EiC for the opportunity of revising the manuscript and submitting a response letter. To ease the review process, we have highlighted all the changes in the manuscript’s revised content in red.

Reviewer1

General feedback:The authors focused on recognizing the typical activities of human beings by using a three-axis accelerator. The work is interesting, but some problems should be clarified and improved:

  1. Comment:The authors should highlight the innovation and contribution of this work.

Response:  We have updated the article. The contributions are now highlighted in abstract and at the end of introduction.

  1. Comment:The introduction of proposed method is not clear. The author should emphasize the unique of proposed method and the reason of its effectiveness rather than the basic knowledge of classification algorithms.

Response: Thank you for spotting this. We have now corrected the introduction of proposed approach. As one contribution of our work is to analyze the effectiveness of each axis of accelerometer, and it is explained in subsection 4.4. The basic knowledge of classification algorithms is given for the readers who don’t work in the field of machine learning.

  1. Comment:The full name and definition of the abbreviations in Eq. (14)-(17) should be given for the readers who don’t work in the field of machine learning.

Response:  Thank you for the suggestion. We have now added the full names of abbreviations and definitions of evaluation measures highlighted in subsection 4.1.

  1. Comment:The explanation of “4.2 Dataset” is very confusing. It seems “Ai” is a specified activity (i.e. the label of the sensor signals). Why is “As” defined by the accelerometer sensor and what is “t”?

Response:  Thank you for spotting this. The confusion is now removed.

  1. Comment:The percentage of data for the training and testing should be clarified. And, the recognition rates in the section of 4.3 is the training result, or the testing result?

Response: Many thanks for the comment. In section 4, we have highlighted the training and testing ratio of dataset. The recognition rate in the section 4.3 is of testing results also mentioned now there.  

  1. Comment:The name of X axis in Fig. 5 should not be “activities”.

Response: Thanks for spotting this. We have corrected the name of x-axis to “classifiers” in Fig. 5.  

  1. Comment:What‘s the unit of the acceleration shown in Fig. 9?

Response: We thank the reviewer for spotting this. We have added the unit of acceleration in Fig. 9.

  1. Comment:The definition of XYZ should be given before discussing the effect of the specified direction for the detected accelerations.

Response: Many thanks for the suggestion. The definition of XYZ is given in subsection 4.3.

  1. Comment:Why could the proposed method have the better result in WISDM dataset?

Response: We thank the reviewer for spotting this. As in axis analysis, we shows that the combination of y-z-axis gives more promising results than the combination of other axis of accelerometer. So, we use only y-z axis of WISDM dataset while other studies used x-y-z axis. Thus, the proposed approach have the better results on WISDM dataset than other studies.

  1. Comment:There are some grammar and typo errors, e.g., the missing of articles and “fro” in L212 of P9.

Response: We can confirm that all the typo error are now removed including the P9.

Reviewer 2 Report

========
General Comments
========

This paper aims to achieve the accurate activity recognition rate by using the only accelerometer installed on the smartphone.
The authors evaluate their method using the open dataset and the dataset collected from the authors.
Since the target activities are core activities in our daily life, this method helps us to be active and healthy.
The proposed method constructed by five steps. (i.e., data collection, preprocessing, feature extraction, data balancing, and classification steps.)

However, the novelty of the proposed method is not explained clearly, as mentioned in the following comments.

========
Major Comments
========

Authors should evaluate their method by the method which trains the learning model without the data from the subjects for the test step, for example, LOSO(Leave-One-Subject-Out) for aligning the condition between other state-of-the-art works.
For example, in the evaluation, which is conducted by [5], the authors use the data from 26 subjects for training and test on the rest ten subjects.
Otherwise, there are works that achieve higher accuracy on the WISDM dataset, like the following.
(1) achieves 94.2% weighted overall accuracy.
(2) achieves 98.09% overall accuracy.

(1) K. K. Oo, "Daily Human Activity Recognition using Adaboost Classifiers on Wisdm Dataset" Published in International Journal of Trend in Scientific Research and Development (ijtsrd), ISSN: 2456-6470, Volume-3 | Issue-6, October 2019, pp.205-209

(2) K.H. Walse, P. V. Dharaskar, V. M. Thakare, "Performance Evaluation of Classifiers on WISDM Dataset for Human Activity Recognition," In Proceedings of the Second International Conference on Information and Communication Technology for Competitive Strategies, March 2016, No. 26, pp.1-7

The authors should explain about the novelty of their method more clearly. 
The data balancing method, like SMOTE and the recognition models (e.g., Decision Tree, Logistic Regression) are generally used in existing works.

Why do the authors not use PCA?
Generally, PCA performs better than cutting off the feature domain like the following work (3).
(3) A. S. A. Sukor, A. Zakaria, and N. A. Rahim, "Activity recognition using accelerometer sensor and machine learning classifiers," 2018 IEEE 14th International Colloquium on Signal Processing & Its Applications (CSPA), Batu Feringghi, 2018, pp. 233-238.

========
Minor Comments
========

In Section 3.4, the authors should reconsider and explain correctly about SMOTE.
The formula generating new sample should be as following.
(s1', s2') = (1, 2) + rand(0, 1) * (3-1, 4-2)

Please reconsider the usage of a sign of inequality.
For example, ( -12 < x > 12 ) in line 231.

In Line 18, "however" is mistyped.

In Line 45 and 46, ")" is not inserted.

In Decision Tree section in Section 3.5, “K” is should be “k”.

Author Response

Reply to Reviewer 2 comments

General feedback:This paper aims to achieve the accurate activity recognition rate by using the only accelerometer installed on the smartphone. The authors evaluate their method using the open dataset and the dataset collected from the authors. Since the target activities are core activities in our daily life, this method helps us to be active and healthy. The proposed method constructed by five steps. (i.e., data collection, preprocessing, feature extraction, data balancing, and classification steps.) However, the novelty of the proposed method is not explained clearly, as mentioned in the following comments:

  1. Comment:Authors should evaluate their method by the method which trains the learning model without the data from the subjects for the test step, for example, LOSO (Leave-One-Subject-Out) for aligning the condition between other state-of-the-art works.

For example, in the evaluation, which is conducted by [5], the authors use the data from 26 subjects for training and test on the rest ten subjects.

Otherwise, there are works that achieve higher accuracy on the WISDM dataset, like the following.

(1) Achieves 94.2% weighted overall accuracy.

(2) Achieves 98.09% overall accuracy.

(1) K. K. Oo, "Daily Human Activity Recognition using Adaboost Classifiers on Wisdm Dataset" Published in International Journal of Trend in Scientific Research and Development (ijtsrd), ISSN: 2456-6470, Volume-3 | Issue-6, October 2019, pp.205-209

(2) K.H. Walse, P. V. Dharaskar, V. M. Thakare, "Performance Evaluation of Classifiers on WISDM Dataset for Human Activity Recognition," In Proceedings of the Second International Conference on Information and Communication Technology for Competitive Strategies, March 2016, No. 26, pp.1-7

Response: We appreciate your comment. As the participants belongs to different age group (21-63). We cannot perform LOSO experiment on our dataset because some participants performed 3 or 4 activities only out of 6 mentioned activities. So, this experiment will show 0% accuracy of some activities which can lead towards low performance overall. Furthermore, we extensively evaluate our approach using 3- fold-cross validation on both self-collected and wisdm dataset.

Although the work (1) and (2) achieved 94% and 98% results but they used a Meta classifier which predicts the activity with the help of 6 classifiers including Random Forest. While our approach achieve 93% results with single fast conventional classifier. Our approach provided fast and accurate recognition of activities.

  1. Comment: The authors should explain about the novelty of their method more clearly.

The data balancing method, like SMOTE and the recognition models (e.g., Decision Tree, Logistic Regression) are generally used in existing works.

Response: Many thanks. We can confirm that the contributions are now highlighted with in abstract as well as at the end of introduction.

  1. Comment:Why do the authors not use PCA?

Generally, PCA performs better than cutting off the feature domain like the following work (3).

(3) A. S. A. Sukor, A. Zakaria, and N. A. Rahim, "Activity recognition using accelerometer sensor and machine learning classifiers," 2018 IEEE 14th International Colloquium on Signal Processing & Its Applications (CSPA), Batu Feringghi, 2018, pp. 233-238.

Response: Thank you for spotting this. We can confirm that we have addressed this comment in section 4.3 above Table 3.

  1. Comment: In Section 3.4, the authors should reconsider and explain correctly about SMOTE.

The formula generating new sample should be as following.

(s1', s2') = (1, 2) + rand (0, 1) * (3-1, 4-2)

Response: Many thanks for spotting this out. We can confirm we have corrected this comment in section 3.4.

  1. Comment: Please reconsider the usage of a sign of inequality.

For example, (-12 < x > 12) in line 231.

Response:  Thanks for noticing this. We have addressed this issue in section 4.3.

  1. Comment: In Line 18, "however" is mistyped.

Response:  We can confirm have addressed this issue.

  1. Comment: In Line 45 and 46, ")" is not inserted.

In Decision Tree section in Section 3.5, “K” is should be “k”.

Response:  We can confirm that we have addressed this issue in section 3.5.

Round 2

Reviewer 2 Report

The reviewer appreciates the authors' efforts to address the reviewer's comments.
However, the followings are not correctly addressed.
[15] tested on ten subjects that were not included in the training data for evaluating the user independent solution.
Generally, the model trained by the subject's data for a test, like the evaluation conducted by the authors, tends to mark a high score.
Therefore, the authors should conduct an evaluation like [15] conducted.
Otherwise, the authors should not compare with such scores of the methods on the same table.

In Section 4.3, the authors use "[-20,20]" and "(-1212)" to show the interval of the value.
It is confusable and should be standardized in the manuscript.
The author should use such mathematical symbols correctly.
"[]" is used to express the close interval.
Is "(-1212)" incorrect use of "(-12<x<12)"?

Author Response

`Revision of Manuscript JManuscript ID: sensors-759417

Analyzing the Effectiveness and Contribution of each Axis of Tri-Axial

Accelerometer Sensor for Accurate Activity Recognition

Submitted to Sensors

We would like to thank the reviewers and Editor-in-Chief for their valuable comments on the manuscript along with their suggestions of improvements. These comments and suggestions have been considered when preparing the revised version of the manuscript. The remainder of this response letter explains how we have handled the reviewers’ comments and implemented their suggestions. We hope that the changes and improvements that we have made in the manuscript will satisfy the requirements of both the reviewers and the Editor-in-Chief, which will lead to accepting the manuscript for publication in a forthcoming issue of Journal of Parallel and Distributed Computing.

Editor in Chief’s initial decision: Major revision

Response:We thank the EiC for the opportunity of revising the manuscript and submitting a response letter. To ease the review process, we have highlighted all the changes in the manuscript’s revised content in red.

Reviewer 2

General feedback:The reviewer appreciates the authors' efforts to address the reviewer's comments.
However, the followings are not correctly addressed:

  1. Comment: [15] tested on ten subjects that were not included in the training data for evaluating the user independent solution. Generally, the model trained by the subject's data for a test, like the evaluation conducted by the authors, tends to mark a high score. Therefore, the authors should conduct an evaluation like [15] conducted. Otherwise, the authors should not compare with such scores of the methods on the same table.

Previous Response:We appreciate your comment. As the participants belongs to different age group (21-63). We cannot perform LOSO experiment on our dataset because some participants performed 3 or 4 activities only out of 6 mentioned activities. So, this experiment will show 0% accuracy of some activities which can lead towards low performance overall. Furthermore, we extensively evaluated our approach using 3-fold-cross validation on both self-collected and WISDM dataset.

New Response:We thank the reviewer for spotting this issue. Although we performed the three-fold cross-validation that works by leaving 1:3 part of dataset for testing and that part does not belong to 2:3 training part ever. This is because, in cross-validation we randomly shuffle the 1:3 and 2:3 parts of dataset for training and testing for 3 times again and again.

For now, we performed the LOSO experiment by our approach on WISDM dataset as [15] performed and compare it with [15] in a separate Table. 5. The explanation is highlighted at the last paragraph of comparative analysis section 4.4.

  1. Comment: In Section 4.3, the authors use "[-20, 20]" and "(-1212)" to show the interval of the value. It is confusable and should be standardized in the manuscript. The author should use such mathematical symbols correctly. "[]" is used to express the close interval. Is "(-1212)" incorrect use of "(-12<x<12)"?

Response: We thank the reviewer for spotting this issue. We have now used closed interval and maintained the consistency in the entire paper.